# GWAS on retinal vasculometry phenotypes

Xiaofan Jiang[1,2], Pirro G. Hysi[2,3,4], Anthony P. Khawaja[5], Omar A. Mahroo[1,4], Zihe Xu[4], Christopher J. Hammond[3,4,6], Paul J. Foster[5], Roshan A. Welikala[7], Sarah A. Barman[7], Peter H. Whincup[8], Alicja R. Rudnicka[8], Christopher G. Owen[8]*, David P. Strachan[8], The UK Biobank Eye and Vision Consortium[¶]

**1** UCL Institute of Ophthalmology, London, United Kingdom, **2** UCL Great Ormond Street Institute of Child Health, University College London, London, United Kingdom, **3** Department of Twin Research and Genetic Epidemiology, King's College London, London, United Kingdom, **4** Section of Ophthalmology, School of Life Course Sciences, King's College London, London, United Kingdom, **5** NIHR Biomedical Research Centre, Moorfields Eye Hospital NHS Foundation Trust and UCL Institute of Ophthalmology, London, United Kingdom, **6** Department of Medical Biochemistry, Oslo University Hospital, Oslo, Norway, **7** Faculty of Science, Engineering and Computing, Kingston University, Penrhyn Road, Kingston upon Thames, Surrey, United Kingdom, **8** Population Health Research Institute, St George's, University of London, London, United Kingdom

¶ Membership of UK Biobank Eye and Vision Consortium is provided in Supporting Information file S1_Acknowledgments.
* cowen@sgul.ac.uk

**Data Availability Statement:** Data are available from UK Biobank through an application process. The process is outlined at the following website: https://www.ukbiobank.ac.uk/enable-your-research/apply-for-access. Other data sources

## Abstract

The eye is the window through which light is transmitted and visual sensory signalling originates. It is also a window through which elements of the cardiovascular and nervous systems can be directly inspected, using ophthalmoscopy or retinal imaging. Measurements of ocular parameters may therefore offer important information on the physiology and homeostasis of these two important systems. Here we report the results of a genetic characterisation of retinal vasculature. Four genome-wide association studies performed on different aspects of retinal vasculometry phenotypes, such as arteriolar and venular tortuosity and width, found significant similarities between retinal vascular characteristics and cardiometabolic health. Our analyses identified 119 different regions of association with traits of retinal vasculature, including 89 loci associated arteriolar tortuosity, the strongest of which was rs35131825 ($p = 2.00 \times 10^{-108}$), 2 loci with arteriolar width (rs12969347, $p = 3.30 \times 10^{-09}$ and rs5442, p = 1.9E-15), 17 other loci associated with venular tortuosity and 11 novel associations with venular width. Our causal inference analyses also found that factors linked to arteriolar tortuosity cause elevated diastolic blood pressure and not vice versa.

## Author summary

Vessels at the back of the eye (the "retina") can be imaged easily. This paper reports on the largest genetic study of retinal vessel shape and size characteristics so far undertaken, to the best of our knowledge. Our study is novel in using an automated artificial intelligence imaging approach to distinguish between arteries and veins, and in demonstrating more genetic associations with vessel characteristics than any previous study (119 genetic loci in all). We also show that the tortuosity of retinal arteries is the most strongly genetically

used include the 1000 Genomes Project (available from http://www.1000genomes.org/), PLINK (https://www.cog-genomics.org/plink2/), the Genotype-Tissue Expression (GTEx) project: (http://www.gtexportal.org/home/), RegulomeDB (http://www.regulomedb.org/), UK Biobank http://www.ukbiobank.ac.uk), MRbase (https://app.mrbase.org/).

**Funding:** The retinal vasculometry work was supported by the Medical Research Council Population and Systems Medicine Board (MR/L02005X/1) and British Heart Foundation (PG/15/101/31889) (PJF,SAB,PHW,ARR,CGO,DPS). PJF has received additional support from the Richard Desmond Charitable Trust (via Fight for Sight) and the Department for Health through the award made by the National Institute for Health Research to Moorfields Eye Hospital and the UCL Institute of Ophthalmology for a Biomedical Research Centre. The views expressed in this article are those of the authors and not necessarily those of the Department for Health. APK is funded by a UK Research and Innovation Future Leaders Fellowship (MR/T040912/1) and an Alcon Young Investigator Award. XJ is funded by Moorfields Eye Charity. OAM is funded by the Wellcome Trust (206619/Z/17/Z). EPIC-Norfolk funding: The EPIC-Norfolk study (DOI 10.22025/2019.10.105.00004) has received funding from the Medical Research Council (MR/N003284/1 and MC-UU_12015/1) and Cancer Research UK (C864/A14136). The clinic for EPIC-Norfolk 3HC was funded by Research into Aging, now known as Age UK (Grant Ref: 262). The genetics work in the EPIC-Norfolk study was funded by the Medical Research Council (MC_PC_13048). We are grateful to all the participants who have been part of the project and to the many members of the study teams at the University of Cambridge who have enabled this research. The funders had no role in study design, data collection and analysis, decision to publish, or preparation of the manuscript.

**Competing interests:** We have read the journal's policy and the authors of this manuscript have the following competing interests: APK has consulted for the following companies: Abbvie, Aerie, Google Health, Novartis, Reichert, Santen and Thea. CJH has consulted for Nevakar Inc.

determined vessel characteristic (replicated remarkable well in a separate second large dataset). In addition, using a particular type of genetic analysis (so called "Mendelian Randomization") we show for the first time that the tortuosity of arteries in the retina is causally related to elevated diastolic blood pressure and not the other way around. This is important as it provides unique insights into the mechanism of elevated blood pressure and hypertension, providing pointers to novel therapeutic targets for future treatment.

## Introduction

The retina is one of the most metabolically active tissues of the human body [1] whose physiological functions require a steady supply of oxygen and nutrients and prompt removal of metabolic waste products. The retina is supplied by branches of the ophthalmic artery, which originates from the internal carotid artery that splits into a central and several ciliary arteries, which penetrate the globe to supply the inner and outer portions of the retina. Retinal vessel calibre, tortuosity and other morphological features vary as a result of constitutive, heritable factors [2], including genes participating in angiogenesis [3, 4], but also as a function of the level of metabolic activity in the retina [5, 6], and systemic haemodynamic changes like blood pressure [7, 8] and blood viscosity [9].

Retinal vessel morphology is an important marker of cardiometabolic [10–12] and eye health [13, 14]. Previously published works have investigated the influence of genetic factors over retinal vessel calibre and tortuosity [3, 4], but, as for many other complex traits, the heritability explained by the identified genes is modest, the underlying mechanisms underlying retinal vessel morphological variations remain poorly characterized, and the specific systemic impairments that each single feature of retinal vasculature reflects, are still poorly understood.

Genome-wide association analyses provide a tool to discover unsuspected common variants having potential risk for complex diseases [15]. There were previously reported genome-wide association studies (GWAS) results on retinal vasculometry characteristics, including venular width (not arteriolar width) [3] and more recently arteriolar and venular tortuosity [4], but sample sizes were small, the number of loci limited (4 at best) [3], and findings from discovery data-sets were not fully replicated [3].

A recent paper has examined the GWAS of retinal vessel density and complexity in a large study of European participants from the UK Biobank identified several loci, but did not distinguish between arterioles and venules, which may have different genetic determinants [16]. Hence, we aim to improve our knowledge of the genetic basis of retinal vasculometry phenotypes in the UK Biobank cohort, for arterioles and venules separately, and the mechanisms that determine their shape and tortuosity, to better understand relationships between the ocular fundus vascular phenotype and systemic conditions.

## Methods

### Ethics statement

UK Biobank was conducted with approval from the North-West Multi-Centre Research Ethics Committee (11/NW/0382), in accordance with the principles of the Declaration of Helsinki and the Research Governance Framework for Health and Social Care. All participants gave written informed consent.

## Participants

The UK Biobank is a large multi-site cohort study of UK residents ages 40 to 69 years who were registered with the National Health Service (NHS) and living up to 25 miles from a study centre. A baseline questionnaire, measurements, and biological samples were undertaken in 22 assessment centres across the UK between 2006 and 2010. All UK Biobank genotypes were obtained as described elsewhere [17].

## Phenotyping and retinal imaging

Ophthalmic assessment was not part of the original baseline assessment and was introduced as an enhancement in 2009 for 6 assessment centres which were spread across the UK (Liverpool and Sheffield in North England, Birmingham in the Midlands, Swansea in Wales, and Croydon and Hounslow in Greater London). Participants completed a touch-screen self-administered questionnaire. Retinal vessel information was extracted from digital fundus photography and spectral domain OCT images. An automated system (the QUARTZ system) can distinguish anatomical features from retinal images, including the optic disc, venules, arterioles and vessel segments, and out-puts centreline coordinates, and measures vessel width and tortuosity [18–20]. QUARTZ measures were summarized using mean width in microns and tortuosity (arbitrary units) [21] weighted by the length of the vessel segment, for arterioles and venules separately for each image, averaged across both eyes to give a person-level mean for analyses. Such an approach was considered appropriate given the pervasive examination of systemic traits.

Image processing modules were all validated on a subset of 4,692 retinal images from a random sample of 2346 UK Biobank participants: modules included vessel segmentation, image quality score, optic disc detection, vessel width measurement, tortuosity measurement, arteriolar venular recognition [18–20]. The performance of the Arteriole/Venule (A/V) recognition had detection rates of up to 96% for arterioles and 98% for venules when the automated probability of arteriole or venule was set to a cut-off of 0.8. An automated assessment of image quality was also made based on the segmented vasculature [18]. The algorithm achieved a sensitivity of 95% and a specificity of 94% for the detection of inadequate images. A model eye was used to quantify the magnification characteristics of the Topcon 3D OCT-1000 Mk 2 fundus camera, allowing pixel dimensions of vessel diameter to be converted to real size [22].

## Statistical analyses

*GWAS*. Details about the UK Biobank study including microarray genotyping and SNP imputation have been described previously [17]. The UK Biobank team performed imputation from a combined panel consisting of the Haplotype Reference Consortium [23] and UK10K reference panels. Phasing on the autosomes was carried out using a modified version of the SHAPEIT2 [24] program.

Ethnicity-stratified GWAS were performed on each of the different retinal vessel phenotypic traits. For this study, we included 52,798 research individuals of European ancestry participating in the UK Biobank. For comparative purposes, additional sets of 933 individuals of South Asian and 1,288 of African ancestry were available and analysed separately (Fig 1). These analyses were underpowered and were used for comparison purposes only, and not for any post-GWAS analyses. Mixed linear regressions, with the retinal vasculometry phenotypes as the dependent variable, the allelic dosage at each locus as the independent predictors, adjusted for age, sex, spherical equivalent and the first ten principal components were conducted using the Bolt-LMM software [25] in the European ancestry subgroup. PLINK [26] was used, after the removal of related subjects, for the analyses of non-European samples. The

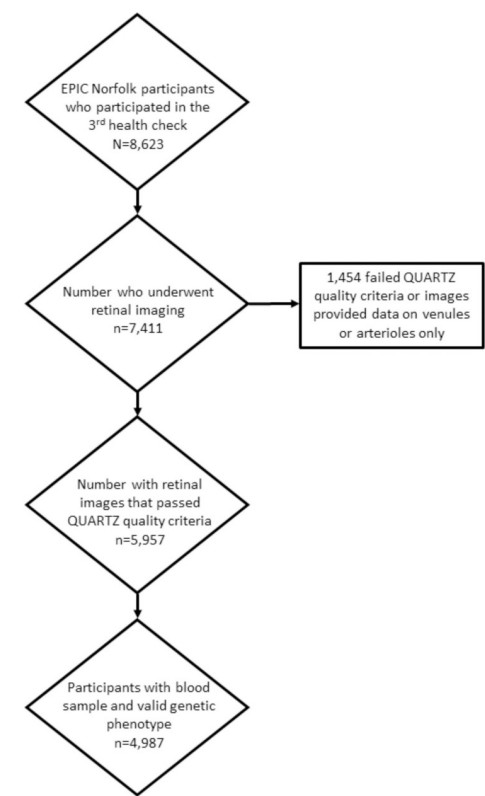

**Participant flow chart for UK Biobank**

UK Biobank participants with retinal images 2009-19 / 2012-13 N=88,052 → 26 withdrew consent and 20,275 failed QUARTZ criteria or images provided data on venules or arterioles only

Number with retinal images that passed QUARTZ quality criteria n=67,751 → 1,425 with baseline repeat images excluded

n=66,326

Participants with blood sample and valid genetic phenotype n=52,798 → For comparison, two ancestries were included. 933 individuals of South Asian, 1,288 of African ancestry

**Participant flow chart for EPIC Norfolk**

EPIC Norfolk participants who participated in the 3rd health check N=8,623

Number who underwent retinal imaging n=7,411 → 1,454 failed QUARTZ quality criteria or images provided data on venules or arterioles only

Number with retinal images that passed QUARTZ quality criteria n=5,957

Participants with blood sample and valid genetic phenotype n=4,987

**Fig 1. Particpant flow diagram.**

GWAS significance threshold was set at the customary $5\times10^{-08}$. Although we conducted four separate GWAS, in part because of the correlation between traits, we did not correct for multiple testing.

Associations are reported for "genomic regions", which we defined as contiguous regions of markers associated at GWAS-level statistical significance separated by less than 1 million base-pairs from each-other. The Online Mendelian Inheritance In Man dataset (www.omim.org) was used to obtain information about diseases caused by rare mutation in any of the genes near our association peaks.

## Replication GWAS—EPIC-Norfolk

The European Prospective Investigation into Cancer (EPIC) study is a pan-European prospective cohort study designed to investigate the aetiology of major chronic diseases [27]. EPIC-Norfolk, one of the UK arms of EPIC, recruited and examined 25,639 participants between 1993 and 1997 for the baseline examination [28]. The EPIC-Norfolk study was carried out following the principles of the Declaration of Helsinki and the Research Governance Framework for Health and Social Care. The study was approved by the Norfolk Local Research Ethics Committee (05/Q0101/191) and East Norfolk and Waveney NHS Research Governance Committee (2005EC07L). Recruitment was via general practices in the city of Norwich and the surrounding small towns and rural areas, and methods have been described in detail previously [29]. Since virtually all residents in the UK are registered with a general practitioner through

the National Health Service, general practice lists serve as population registers. All participants gave written, informed consent. Ophthalmic assessment formed part of the third health examination and this has been termed the EPIC-Norfolk Eye Study [30]. In total, 8,623 participants were seen for the Eye Study between 2004 and 2011. Although de-anonymising UK Biobank participants or linking them with external datasets is not allowed, we expect no overlap between discovery and replication datasets, because the EPIC-Norfolk study participants were recruited in an area geographically distinct from where the UK Biobank participants were recruited. Digital photographs of the optic disc and macula were taken using a TRC-NW6S non-mydriatic retinal camera and IMAGEnet Telemedicine System (Topcon Corporation, Tokyo, Japan) with a 10-megapixel Nikon D80 camera (Nikon Corporation, Tokyo, Japan). Pupils were not dilated. All participants gave written, informed consent.

Genotyping was carried out using the Affymetrix UK Biobank Axiom Array (the same array as used in UK Biobank). SNP exclusion criteria included: call rate < 95%, abnormal cluster pattern on visual inspection, plate batch effect evident by significant variation in minor allele frequency, and/or Hardy-Weinberg equilibrium $P < 10^{-7}$. Sample exclusion criteria included: DishQC < 0.82 (poor fluorescence signal contrast), sex discordance, sample call rate < 97%, heterozygosity outliers (calculated separately for SNPs with minor allele frequency >1% and <1%), rare allele count outlier, and impossible identity-by-descent values. We removed individuals with relatedness corresponding to third-degree relatives or closer across all genotyped participants. 99.7% of EPIC-Norfolk are of European descent and we excluded non-White participants. Imputation was carried out using the Sanger imputation service (https://imputation.sanger.ac.uk) with reference to the Haplotype Reference Consortium panel, version 1.

## Linkage Disequilibrium (LD) score regression analyses

Bivariate regression intercepts were calculated to distinguish between polygenetic effects and effects of population stratification within and between GWAS. Inter-trait genetic correlation on the summary statistics from GWASs was performed using LD Hub [31] and the LD score regression program for phenotypes that were not available through the hub [32]. All the parameters used for the LDSC analyses were following the authors' recommendations [31] for both the analyses using the LD Hub GWAS information and our own.

## Tissue enrichment across multiple tissues

Multi-tissue expression enrichment analyses were performed in order to identify the expression levels in particular tissues using LD-score regression (LDSC) based procedures described elsewhere [33]. The GWAS results were compared to gene expression data assembled and made available by the authors of the LDSC program (https://storage.googleapis.com/broad-alkesgroup-public/LDSCORE/LDSC_SEG_ldscores/Multi_tissue_gene_expr_1000Gv3_ldscores.tgz).

## Transcriptome-based association analyses

We applied whole-genome transcriptomic prediction models trained with reference data from version 8 of the Genotype-Tissue Expression (GTEx) project [34] to infer transcriptome variation in the GWAS study and then compute transcriptome-to-trait associations using GWAS associations' summary statistics as input. We used the S-PrediXcan programme [35, 36] to estimate genetically regulated gene expression using whole-genome tissue-dependent prediction models, trained on GTEx version 8 reference transcriptome data and using GWAS-generated information to infer gene-based associations with retinal vasculature phenotypes. Based

on the GTEx analysis described above, we applied this framework to 49 GTEx tissues and 4 retinal vasculometry traits.

### Integration of genotypic and expression data

Summary-based Mendelian Randomization (SMR) integrates summary-level GWAS data with expression quantitative trait loci (eQTL) studies, to identify pleiotropic associations between a complex trait and a specific gene [37].

Due to availability constraints and the need to minimise the number of tests, the eQTL-based analyses were performed in selected tissues only. We selected tissues where the original sample size in the GTEx datasets was comparatively large. To maximise power, we conducted these analyses in subcutaneous adipose (N = 581), tibial artery (N = 584, chosen a priori in preference to coronary artery due to increased sample size), and a less specific dataset of less-tissue specific leukocytes for which larger sample sizes (N = 5,311) were available [38]. Tests based on cis-methylation information (cis-mQTL), were carried out in brain tissues [39].

### Mendelian Randomisation (MR)

The MR-base package [40] was used for two sample MR analyses testing for causality of vascular changes over metabolic traits. MR-base uses instruments SNPs selected on the basis of several hundreds of GWAS summary data sets in its repository and estimates the causal impact of specific traits (exposures) on retinal vasculature phenotype outcomes. These SNPs are all significantly associated with the "exposure" trait and they are meant to be in linkage equilibrium with each other, even if they are located in relative proximity of each other. In all cases, our MR cases were two-sample; the genetic effects were obtained from our GWAS on the four retinal vasculometry traits of interest, and they were compared with effect estimates on other traits from samples that did not include any UK Biobank subjects. Studies in which the UK Biobank had contributed information were specifically removed from the analyses reported here. We are reporting results from different MR tests: the inverse variance weighted, weighted median and MR-Egger tests (both MR and intercept). The results are primarily reported with the reference of the random-effect inverse variance weighted test, but the other tests may be valuable to interpret the relationship between exposure and outcome. In particular, the MR-Egger intercept tests for unbalanced (unidirectional) horizontal pleiotropy, which is usually taken as evidence against causation.

## Results

### Association results

We performed four separate GWAS of arteriolar tortuosity (AT), arteriolar width (AW), venular tortuosity (VT) and venular width (VW) in a total of 52,798 participants of European ancestry from the UK Biobank. The GWAS results were all in line with polygenic architecture for all these traits, without any sign of inflation due to uncontrolled population structure (LDSC intercept in the range 1.01–1.04, Table A in S1 Table). The four retinal vasculature traits examined were genetically correlated with each-other, with the highest correlation ($r_g$ = 0.44) observed between VW and AW (Table B in S1 Table), suggesting that these parameters estimate complementary aspects of the retinal vasculature.

The trait for which SNPs explained the largest proportion of variance (SNP heritability) was AT ($h^2_{SNP}$ = 0.51). We also observed some of the most significant associations between this trait and polymorphic variants across the genome. A total of 14,021 SNPs were associated at genome-wide level with AT, and they clustered around 89 different unique genomic regions

(Table C and Fig A i-iv in S1 Text). The most statistically significant association was observed for polymorphic variants within the genomic sequence of the *COL4A2* gene ($p$ = 2.0x10$^{-108}$ for rs35131825). Other novel significant associations with this trait were observed for polymorphisms overlapping with the genomic sequences of genes involved in cardiovascular function, such as those within the *PDE3A* gene ($p$ = 1.30x10$^{-64}$ for rs11045245). The statistically strongest association for VW was found in the region located on chromosome 6q24 ($p$ = 1.40x10$^{-14}$ for rs12206319), within the *NMBR* gene. The strongest association for AW was identified with a SNP (rs5442, $p$ = 1.9x10$^{-15}$) within *GNB3* gene [41]. The strongest significant association for VT was found for a SNP (rs1136956, $p$ = 6.0x10$^{-58}$) within the protein coding region of the *ACTN4* gene. Interestingly, this region was also significantly associated with all retinal vasculometry phenotypic traits that we analysed, except for AW.

In addition to the genetic loci above, which have been previously related to vascular morphology features, we observed unique associations that have not been previously associated with vascular, or any other phenotypic trait. For example, strong association with AT was found for rs1950127 ($p$ = 1.80×10$^{-68}$), located within the coding sequence of LOC101928978, a transcript of unknown function. Novel association with AW was found for one gene locus (*CLUL1*, $p$ = 3.30×10$^{-09}$ for rs12969347). In addition, a novel association with VT was found, among others, at a locus overlapping with the genomic sequence of the *DLX6* gene ($p$ = 4.30 × 10$^{-23}$ for rs2948244), a transcriptor gene linked to forebrain and craniofacial development.

## Replication and transethnic comparison

We sought to replicate these findings in a fully independent population sample. For this purpose, we used results obtained from the EPIC-Norfolk study, a population-based cohort whose general ethnic and demographic characteristics as well as ophthalmic assessment modalities closely matched those of the UK Biobank participants. Despite the considerably smaller sample size replication was robust for all traits; for example, SNPs associated with arterial tortuosity in the UK Biobank were associated at a Bonferroni ($p$ < 0.05/73 available SNPs) over 11 loci (Table D in S1 Table) and there was generally a linear relationship between effect sizes estimated in the discovery and replication cohorts (Fig B in S1 Text).

We subsequently meta-analysed data from both cohorts and, as expected, we obtained associations not previously seen in the analyses of the UK Biobank data only (Table E in S1 Table). However, given that we were unable to provide further replication to these new results, in the following sections of the manuscript, we will continue to refer to results obtained from the UK Biobank only.

Next, we sought to investigate the variation in the amplitude of the association signals across the different available ancestral groups (Table F in S1 Table), by comparing results from the European panel with association data from 1,288 African and 933 South Asian UK Biobank participants. Due to sample size differentials, p-value based comparisons were underpowered. However, we observed considerable correlations between effect sizes observed in subjects of European and African (Pearson's r = 0.33, 95% CI [0.12, 0.51], p = 0.02]) and South Asian ancestry (r = 0.46, 95% CI [0.27, 0.61], p<0.00001). Fig C in S1 Text shows the correlated relationship between different ancestry.

## Tissue expression enrichment and exploration of functional mechanisms in vasculometry regulation

Analyses of eQTL offer insights on the genetic architecture of expression regulation. These analyses also help to annotate the exact genes among the many located within associated

regions whose variations are at the origin of the association signals. Many of the GWAS associations were expressed across a wide range of tissues (Table G in S1 Table). Due to considerable tissue variability of expression patterns, we first sought to identify the tissues where our GWAS results observed in European participants were most enriched. A multi-tissue enrichment analysis for traits [33] found that, among available GTEx tissues/cell types, genes associated with retinal vessel parameters in our four GWAS analyses showed statistically significant enrichment in adipose tissues (Fig D in S1 Text), but also in artery, venous tissues, as well as smooth muscle-rich tissues such as oesophagus and myometrium tissues. Genes associated with VW diverged most from the pattern observed for the other vasculometry traits and were mostly expressed in kidney and pancreas, although the results of these analyses were not statistically significant after multiple testing correction.

We subsequently tested the hypotheses that many of our GWAS findings were exerting their effects over the phenotypes through the eQTL effects affecting the transcription of the genes. Using tissues in which we observed the highest enrichment of GWAS results (subcutaneous adipose and tibial artery) as well as white blood cells due to the abundance of information available for them, we conducted summary-based Mendelian randomization tests for loci associated with AT, which was the GWAS analysis for which we obtained most results and therefore had most power (Fig E in S1 Text; Table A in S1 Table shows AT heritability value 0.512). For the other retinal vasculature traits, the SNP instruments were of low power due to relatively few associations, meaning non-significant results in MR causality analyses. We found statistically significant ($p_{\text{HEIDI}} \geq 0.05$) evidence that many SNPs associated with AT causally influences this phenotype through the control of the transcription process of several genes, such as the SNPs in the *ACTN4* locus (Table H in S1 Table), which was associated with three out of the four retinal vessel phenotypes we analysed in our dataset.

## Relationship of vasculometry and other phenotypes

To establish a relationship between the retinal vessel phenotypes that we analysed, with other human phenotypic traits or disease, we calculated the genome-wide correlation between effect sizes observed in our GWAS with those estimated for other diseases. These analyses suggest that our phenotypes share significant proportions of their genetic risk with a number of cardiometabolic traits and disease (Table I in S1 Table). Statistically significant genetic effects' correlations were observed, notably between retinal vessel parameters and metabolic phenotypes such as the systolic and diastolic blood pressure, cardiovascular disease, fat mass, but also standing body height, etc.

We consider retinal vasculometry parameters as biomarkers of vascular changes elsewhere in the body. We further explored causality by performing two-sample MR analyses of potentially relevant traits that were significantly correlated with the retinal vasculometry phenotypes that we analysed (Table J in S1 Table). For these purposes, we used data that were available through the *MR-base* platform. We specifically tested the hypotheses that processes associated with retinal vasculometry variability can lead to other diseases. We found statistically significant evidence that processes underlying arteriolar tortuosity in the retina cause elevated diastolic blood pressure ($p_{\text{IVW}} = 3.5 \times 10^{-05}$, Fig 2). Our analyses also suggested that changes in arteriolar tortuosity may also causally affect risk of coronary heart disease.

We also tested the reverse model that these phenotypes causally lead to changes in arterial tortuosity, using the limited analyses on whose traits whose summary statistics are publicly accessible (Table K in S1 Table). We did not find any evidence of reverse causation, which further reinforces the confidence in the original MR findings that identified AT as one of the causes for elevated diastolic blood pressure and a potential contributing cause to coronary heart disease.

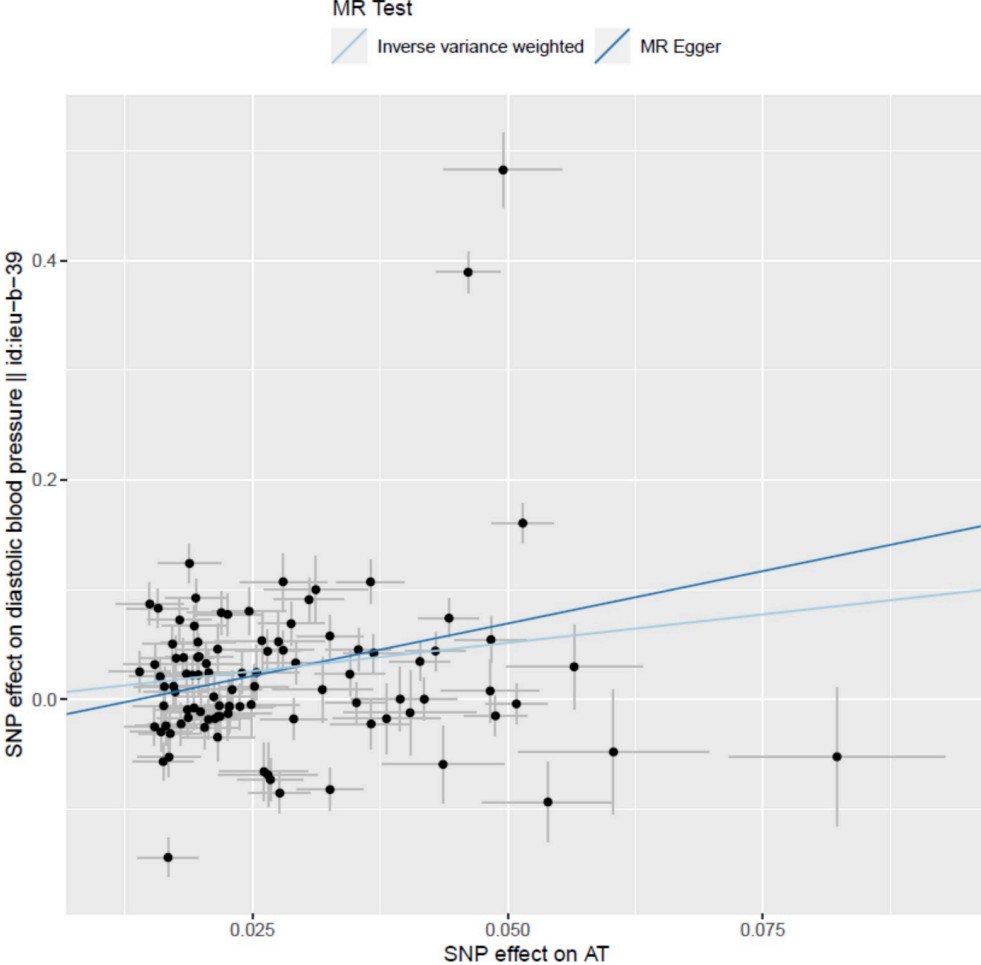

**Fig 2. Results of Mendelian Randomization analyses for arteriolar tortuosity (AT) as a causal influence on diastolic blood pressure (DBP).** Footnotes: Among 108 SNPs that showed independent genome-wide significant associations with AT in conditional regression analysis, 99 had published information available on their association with DBP in non-UK-Biobank studies [reference: decipher "id:ieu-b-39" and remove from y-axis label]. Single points in the graph represent co-ordinates determined by the effect of each specific SNP on DBP (in non-Biobank studies) and AT (in UK Biobank). Outlying SNPs are identified individually by rs-number. [see comment below.]. MR analyses were conducted with all 99 SNPs as instrumental variables using the conventional inverse-variance-weighted (IVW) and Egger methods. The slope of each line corresponds to the estimated MR effect for each method. Significance tests for the MR slope (Table I in S1 Table) were: p = 0.00004 (3.55E-05) by MR-IVW. p = 0.0050 (0.005017737) by MR-Egger. Using a weighted-median MR method (which is less sensitive to outliers), the corresponding p-value was p = 0.00001 (1.01E-05).

## Discussion

Based on the largest dataset of retinal vessel morphology to date, we identified 119 different loci associated with retinal vasculometry traits. We found the most loci (n = 89) associated with AT, and 2 novel loci associated with AW, 17 loci associated with VT, and 11 associations with VW. Our study used AI-enabled automated image processing techniques to derive these retinal vasculometry traits, which have been developed, validated internally and externally, and applied in a harmonised fashion across two large population-based cohorts, namely UK Biobank and EPIC-Norfolk cohorts [11, 12, 18, 20, 21]. Previous work using our AI-enabled automated retinal vessel image analysis system (QUARTZ) has described phenotypic associations of these four vasculometry indices (i.e., AW, AT, VW, VT) with conventional

cardiometabolic risk factors including blood pressure (BP) [11], adiposity/BMI [12], DM and blood lipids [21]. Analysis in UK Biobank has demonstrated strong associations between VW and VT with markers of adiposity [12], and that AW, VW and AT all show strong inverse associations with blood pressure (systolic and diastolic) [11]. Hence, these microvascular characteristics could be the cause or consequence of important pathophysiological traits. We sought to clarify the direction of causality by examining their genetic determinants by instrumental variable (Mendelian randomization, MR) analysis.

Several loci have been identified by earlier, statistically less-powered studies with smaller sample sizes. A meta-analytic study of over 18,000 participants of European ancestry identified rs2194025 locus on chromosome 5q14 near the *MEF2C* gene as being associated with retinal arteriolar diameter [42], later updated to include over 24,000 multi-ethnic participants to find 4 loci associated with both central retinal vein and arterial width [41]. We were unable to replicate these findings, with our most significant associations for AW being on chromosomes 18 and 22, with completely different loci (Table E in S1 Table). More recent studies have focussed on examining genetic determinants of retinal vessel tortuosity. A study of just over 3000 individuals of European ancestry showed one locus associated with VT (*ACTN4/CAPN12* genes), and another associated with AT (*COL4A2* gene) [4]. A recent study using the UK Biobank data source in fewer participants, showed that retinal vessel density and fractal dimensions, as a measure of vessel complexity (extracted from the entire image after deep learning vessel segmentation without distinction between arterioles and venules) were associated with 7 and 13 novel loci respectively [16]. The fewer number of loci identified, suggesting a lower level of heritability, may reflect the indiscriminate identification of vessels (without being able to distinguish between arterioles and venules), and while the strong signal between overall vessel density with MEF2C was replicated by the association we observed specifically with venular width, the level of overlap with our vasculometry phenotypes was limited to 8 loci (Table L in S1 Table), and the strong signal with OCA2 was not observed.

In our study, the most statistically significant association with AT was observed for polymorphic variants within the genomic sequence of the *COL4A2* gene ($p = 2.0 \times 10^{-108}$ for rs35131825). Importantly, these associations were apparent in both the validation UK Biobank dataset (also shown in a sub-set of non-European ancestry, see Tables D and E in S1 Table), and the EPIC-Norfolk replication dataset with remarkable replication. The *COL4A2* gene is implicated in angiogenesis and tumour growth suppression [43] and previous work has reported specific changes in its genomic sequence that cause familial and sporadic small vessel disease [44, 45]. Interestingly, the COL4A2 gene is located next to the COL4A1 gene on chromosome 13q34. COL4A1 is known to harbour mutations that cause familial retinal arteriolar tortuosity [46]. It is therefore unclear if COL4A1, COL4A2 or both can contribute to changes in the retinal vasculature. Other novel significant associations with this AT trait were observed for polymorphisms overlapping with the genomic sequences of genes involved in cardiovascular function, such as those within the *PDE3A* gene ($p = 1.30 \times 10^{-64}$ for rs11045245), a gene previously associated with the size of the aortic root [47] and blood pressure [48], but to the best of our knowledge, not with retinal or other small vessel morphological features.

Significant associations with loci that have been previously implicated in angiogenesis and regulation of vasculature function was a feature of all four traits we examined. For example, the statistically strongest association for VW was found in the region located on chromosome 6q24 ($p = 1.40 \times 10^{-14}$ for rs12206319), within the *NMBR* gene, which has previously been associated with retinal vascular calibre [3]. The strongest association for AW was identified for a SNP (rs5442, $p = 1.9 \times 10^{-15}$) within the *GNB3*, a gene also associated previously with central retinal vein calibre [41], but also multiple ocular phenotypes such as refractive error [49], macular thickness [50] and corneal astigmatism [51]. The strongest significant association for VT was

found for a SNP (rs1136956, $p = 6.0 \times 10^{-58}$) within the protein coding region of the *ACTN4* gene. The *ACTN4* gene codes for an alpha actinin protein, a ubiquitous and multifunctional family involved in the cytoskeleton framework and cell-cell adhesion [52]. Interestingly, this region was also significantly associated with all retinal vasculometry phenotypic traits that we analysed, except for AW.

Another important novel question we sought to answer from this study was the relationship between retinal vasculometry and systemic metabolic traits. This is particularly relevant for BP, in examining whether microvascular changes (as evidenced in the retina in our study) precede or cause raised BP, or whether they result in hypertension. Genetic influences on AT were strongest with high heritability ($h^2$ value 0.5), and intermediate for VW and VT (with $h^2$ values ~0.2), offering scope for the MR approach to be used for all these three traits. The low heritability observed for AW deprived us of the opportunity to explore the systemic consequences of factors linked to AW variation. These differences suggest that these vasculometry traits might be differently genetically and phenotypically determined, with AT being by far the most heritable of the 4 traits, exhibiting associations with 89 unique genomic regions. Using Mendelian Randomization (MR), we provide strong statistical evidence ($p = 3.5 \times 10^{-5}$) that processes leading to increased AT causally influence blood pressure.

To explore the direction of causality we took advantage of the availability of powerful "instrumental variables" drawn from among genetic variants strongly predisposing to traits of interest. However, the SNPs with the largest per-allele effects on AT had little or no effect on diastolic BP (Fig 2), suggesting that there may be multiple biological pathways represented among the 99 SNPs that were independently associated with AT and selected for our MR analysis, some not implicated in DBP. It should also be noted that although we find evidence to support a causal influence of (genetically determined) AT on BP (particularly diastolic), AT is not the most important trait observationally correlating with DBP (stronger phenotypic associations with DBP are observed for AW than for AT [11]). So, if the AT-DBP association is causal, it is clearly not the whole story as far as microvasculature and systemic blood pressure is concerned. It is noteworthy, that while a recent study concluded that polygenic risk scores for hypertension and T2 diabetes might cause retinal vessel changes (specifically vessel density and fractal dimensions), unlike our study there were too few specific genetic markers to confirm the direction of causality [16].

The major strengths of our study lie in the very large sample sizes, standardised and validated trait measures [11, 12, 18, 20, 21], comprehensive genomic data including imputations, replication of GWAS results, and interpretation of our results both biologically (by use of quantitative trait locus analyses) and epidemiologically (by use of MR approaches). Although our primary analyses focused on European ancestry populations, we considered other ancestry groups in sensitivity analyses. The main limitation we encountered was the lack of genetic associations with AW and the modest number of loci associated with VW and VT, all traits with a low SNP-based heritability, which limited the statistical power of MR analyses of these venular traits. Also, while the use of person level averages in vasculometry phenotypes was suitable to examine systemic traits, it was not possible to examine ocular outcomes that could have led to between-eye vessel asymmetry (e.g., Central Retinal Vein Occlusion), and potentially heritable conditions leading to ocular media opacity (such as corneal disease and cataract), which may have impacted on retinal image quality. However, this affect is likely to be modest given that only a portion of the vascular tree needed to be imaged (as opposed to the whole image), resulting in high inclusion rates (~80% in both UK Biobank and EPIC-Norfolk cohorts). Another potential issue is the need for adjustment for covariates, given that SNPs associated with unadjusted covariates could generate spurious signals. However, this is of practical importance only when there are strong correlations with both the dependent and

independent variables. Theoretically, covariate-adjustment may reduce the residual variance of the outcome and thereby increases power to detect covariate-independent genetic associations. For this reason, we have run models for traits of interest with additional adjustment for age (and age-squared to examine presence of non-linearity), diabetes mellitus and current smoking status, which reassuringly appears to have little effect (Table M in S1 Table).

In conclusion, microvascular morphology, measured objectively and systematically in retinal images, shows distinct phenotypic traits of arterial and venular width and tortuosity, each with distinct patterns of genetic association. For arterial tortuosity, supportive evidence emerges for microvascular morphology as a contributory cause, rather than a consequence, of raised blood pressure, raising the possibility of novel treatment pathways.

## Supporting information

**S1 Acknowledgments. UK Biobank Eye and Vision Consortium.**
(DOCX)

**S1 Text. Supplementary Figures.** Fig A. Manhattan plots depicting the genome-wide associations for arteriolar tortuosity (i), arteriolar width (ii), venular tortuosity (iii) and venular width (iv). Fig B. The relationship between effects sizes estimated in association with retinal vessel parameters in the UK Biobank (x-axes) and those estimated in the EPIC-Norfolk cohort (y-axes). Fig C. The relationship between effects sizes estimated in association with retinal vessel parameters in the UK Biobank (x-axes) and those estimated in the Africa/Asian ancestry (y-axes). Fig D. Tissue expression enrichment analysis results, for AT (i), AW (ii), VT (ii) and VW (iv). Fig E. SMR Integration of results combined GWAS and eQTL for AT on Peripheral blood.
(DOCX)

**S1 Table. Supplementary Tables.** Table A. GWAS quality control metrics. Table B: Genetic intercorrelations between the four retinal vasculature parameters. Table C. Most significant associations observed for the traits of interest. Table D. Replication of GWAS results in the EPIC-Norfolk cohort. Table E. Results of the meta-analyses of the UK Biobank and EPIC-Norfolk cohorts. Table F. Known loci replicated in trans-ancestry analysis combining Asian, African and European analysis. Table G. eQTL effects of SNPs associated in the GWAS. Table H. Functional analyses of causation for SNPs associated with AT. Table I: Ldsc genetic correlation between retinal parameters and systemic traits and disease. Table J. Mendelian randomisation analyses testing causality of vascular changes over metabolic traits. Table K. Mendelian randomisation analyses testing causality of metabolic traits over vascular changes. Table L. Replication of GWAS results in a recent publication (Zekavat BS et al., 2021). Table M. Comparison of part associations observed for the traits of interest with additional covariates (age2, smoking, diabetes).
(XLSX)

## Author Contributions

**Conceptualization:** Pirro G. Hysi, Anthony P. Khawaja, Alicja R. Rudnicka, Christopher G. Owen, David P. Strachan.

**Data curation:** Xiaofan Jiang, Pirro G. Hysi, Anthony P. Khawaja, Zihe Xu, Paul J. Foster, Roshan A. Welikala, Sarah A. Barman, Alicja R. Rudnicka, Christopher G. Owen, David P. Strachan.

**Formal analysis:** Xiaofan Jiang, Pirro G. Hysi, Anthony P. Khawaja, Zihe Xu, Roshan A. Welikala, Sarah A. Barman, Alicja R. Rudnicka, David P. Strachan.

**Funding acquisition:** Pirro G. Hysi, Anthony P. Khawaja, Christopher J. Hammond, Paul J. Foster, Sarah A. Barman, Peter H. Whincup, Alicja R. Rudnicka, Christopher G. Owen, David P. Strachan.

**Investigation:** Xiaofan Jiang, Pirro G. Hysi, Anthony P. Khawaja, Omar A. Mahroo, Zihe Xu, Christopher J. Hammond, Paul J. Foster, Roshan A. Welikala, Sarah A. Barman, Peter H. Whincup, Alicja R. Rudnicka, Christopher G. Owen, David P. Strachan.

**Methodology:** Xiaofan Jiang, Pirro G. Hysi, Anthony P. Khawaja, Omar A. Mahroo, Zihe Xu, Christopher J. Hammond, Paul J. Foster, Roshan A. Welikala, Sarah A. Barman, Peter H. Whincup, Alicja R. Rudnicka, Christopher G. Owen, David P. Strachan.

**Resources:** Pirro G. Hysi, Anthony P. Khawaja, Christopher J. Hammond, Paul J. Foster, Roshan A. Welikala, Sarah A. Barman, Alicja R. Rudnicka, Christopher G. Owen, David P. Strachan.

**Software:** Roshan A. Welikala, Sarah A. Barman.

**Supervision:** Pirro G. Hysi, Sarah A. Barman, Alicja R. Rudnicka, Christopher G. Owen, David P. Strachan.

**Validation:** Xiaofan Jiang, Pirro G. Hysi, Anthony P. Khawaja, Omar A. Mahroo, Zihe Xu, Christopher J. Hammond, Paul J. Foster, Roshan A. Welikala, Sarah A. Barman, Peter H. Whincup, Alicja R. Rudnicka, Christopher G. Owen, David P. Strachan.

**Visualization:** Xiaofan Jiang, Pirro G. Hysi, Anthony P. Khawaja, Omar A. Mahroo, Zihe Xu, Christopher J. Hammond, Paul J. Foster, Roshan A. Welikala, Sarah A. Barman, Peter H. Whincup, Alicja R. Rudnicka, Christopher G. Owen, David P. Strachan.

**Writing – original draft:** Xiaofan Jiang, Pirro G. Hysi, Christopher G. Owen, David P. Strachan.

**Writing – review & editing:** Xiaofan Jiang, Pirro G. Hysi, Anthony P. Khawaja, Omar A. Mahroo, Zihe Xu, Christopher J. Hammond, Paul J. Foster, Roshan A. Welikala, Sarah A. Barman, Peter H. Whincup, Alicja R. Rudnicka, Christopher G. Owen, David P. Strachan.

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
