## [Decision Letter · Decision Letter 0]

30 Aug 2022

Dear Dr Owen,

Thank you very much for submitting your Research Article entitled 'GWAS on retinal vasculometry phenotypes' to PLOS Genetics.

The manuscript was fully evaluated at the editorial level and by independent peer reviewers. The reviewers appreciated the attention to an important problem, but raised some substantial concerns about the current manuscript. Based on the reviews, we will not be able to accept this version of the manuscript, but we would be willing to review a much-revised version. We cannot, of course, promise publication at that time.

If you decide to revise the manuscript for further consideration at PLOS Genetics, please aim to resubmit within the next 60 days, unless it will take extra time to address the concerns of the reviewers, in which case we would appreciate an expected resubmission date by email to plosgenetics@plos.org.

[LINK]

We are sorry that we cannot be more positive about your manuscript at this stage. Please do not hesitate to contact us if you have any concerns or questions.

Yours sincerely,

Sudha K Iyengar

Academic Editor

PLOS Genetics

Hua Tang

Section Editor

PLOS Genetics

Reviewer's Responses to Questions

**Comments to the Authors:**

Reviewer #1: The manuscript by Jiang et al describes genome-wide analyses of retinal vascular parameters with several bioinformatic analyses designed to identify phenotypic overlap. The manuscript is largely well-written and fairly clear, however there are some elements which would benefit from additional clarification. Broadly speaking, it would add quite a bit to the manuscript to discuss the results in the context of gene regulation across your various tissues in addition to phenotypic causation. Other comments include:

1. Is there overlap between EPIC-Norfolk and the UKB participants included in this study? Given the similar catchment areas it would be best to include a statement of independent samples.

2. Were the parameters similar for the LDSC and LD Hub analyses? Specifically LDSC recommends filtration to HapMap SNPs, and use of stringent info score thresholds, were identical thresholds applied in both approaches?

3. Which version of GTEx was used for your S-PrediXcan analysis? 44 tissues would suggest use of v6 rather than the newer (and larger) v7 or v8, and if so ought to be stated. Also where are the results from these analyses? They're never discussed or presented.

4. The tissue enrichment methods should be described in bit more detail, specifically indications of the versions of databases used, etc. as the 200+ tissues interrogated are not listed in more detail. The results indicates GTEx tissues were used for enrichment, but that doesn't add up - there are not 200+ tissues available in GTEx, and myometrium isn't one of them. Please clarify.

5. I am puzzled by the selection of tibial artery rather than coronary artery as one of the tissues, given that coronary artery was a top signal in two of the four analyses.

6. The manhattan plots show many significant signals, far more than are mentioned in the case of arteriolar width. If filters were applied, that needs to be reflected in what is plotted and described more thoroughly. If not, then the results are not described properly and there would be more than enough datapoints to enable comparison of effect sizes between EPIC-Norfolk and the UKB.

7. This isn't critical, but it would be nice to see figures analogous to Supplementary Figure 2 for the cross-ancestry analyses in addition to the supplementary table. This information should also be mentioned in the discussion paragraphs of particular gene results.

8. Line 278: hereditability = heritability

9. The cross-disease genetic correlation analyses seem highly unfocused. The numbers of phenotypes need to be clarified, preferably restricted to those with a scientific rationale ("Distance between home and job workplace"). Also ST9 lists all populations as UKBB, of which your discovery analyses is a subset, this is somewhat concerning.

10. Previously identified results from reference 42 are mentioned in the second paragraph of the discussion, were these among the top hits in this manuscript, and/or were these SNPs evaluated systematically? Noting the replication of these loci would be useful.

Reviewer #2: The authors perform a large GWAS using four QUARTZ-derived fundus features as quantitative traits in 52,798 UK biobank individuals. While this is not the first GWAS on vascular features of the UK biobank fundus photos, this is an important topic of investigation given the link to systemic disease. The research methods in this paper are sound. I do think there needs to be substantial additional consideration to the analysis that I will discuss below. I have some general comments followed by more specific page/line comments:

Image processing: I do not have experience with QUARTZ but it seems reasonable. However, I think that the description of imaging processing needs more detail (I looked in the supplemental but do not see any further detail):

• The authors do not describe how they handle each eye – is one eye used? Both eyes? If both, how do they account for correlation? How do they handle significant differences between the two eyes and the possible confounding etiologies?

• How do the authors correct for image quality? This will degrade with heritable things like corneal disease and cataract, and I assume degradation of the image will systematically affect the 4 measurements.

• How about correcting for retinal pathology in general? Are these all patients without a diagnosis of certain major categories of ocular disease? For example: someone with a history of retinal laser (for diabetes, a retinal tear, a retinal detachment, etc) will have different vasculometry measurements due to the ablation of the retina in the area of laser. Traits leading to the need for laser are heritable.

The GWAS itself: the methods are sound and use well-established analysis pipelines. The authors correct for age, sex, spherical equivalent and the first ten principal components. My main concerns revolve around confounders and clear delineation about findings here that imply a novel pathway as separate from one that is already well understood by ophthalmologists. To be specific:

• Smoking status is not controlled for in this paper – it should be added as a covariate, as it has definite secondary effects on the vasculature throughout the body.

• retinal vein occlusion (or impending vein occlusion), one of the more common secondary reasons for vascular tortuosity, is not mentioned in this paper. A sign of this would be a difference in tortuosity between the two eyes (in the case of CRVO and assuming it wasn’t bilateral) or a sectoral change within the eye (BRVO). Can the authors account for this in any way (there are likely several ways of doing this)?

• Systemic hypertension, cardiovascular disease, diabetes and (to a certain degree) hyperlipidemia are other well known clinical associations with vascular tortuosity on exam. In a way, it is not novel to find SNPs that co-associate with VT and these traits. Yet, there is no analysis here that controls for this. One could argue that controlling for these traits is not necessarily desired, because they are in the causal pathway, but it would be interesting to have a secondary analysis that controls for blood pressure and diabetes (for example) and reports SNPs that associate independent of this confounder that is already understood as associated with vascular tortuosity.

• It is standard to correct for age-squared to model its effect and account more accurately for any non-linearity

• Uncontrolled diabetes and hypertension (amongst other things) can lead to vascular narrowing and pruning and eventual non-perfusion of the peripheral retina. I assume the time with respect to disease onset will have an effect on the vasculometry measurements. Can the authors discuss how they handled this (or why they decided not to control for it)?

• I don’t believe genotyping array is added as a covariate and it likely should be if it varied in this large sample size

More specific comments:

Page 3 line 68: This description of the retinal vasculature is not correct. The long and short ciliary artery split off BEFORE penetrating the sclera. Together, the long posterior ciliary artery and the short posterior ciliary arteries feed the choroid (long and short), iris (long) and ciliary body (long).

Page 3 line 73: Retinal vein occlusion (or impending vein occlusion) should have been considered in this manuscript as a cause for venous tortuosity, especially if unilateral.

Page 3 line 74: Cite Zekavat et al. Circulation 2022 as a paper that is already published using the UK biobank to study retinal vasculature. Please alter statements about novelty and explain the distinction between VT/VD and the traits in the Zekavat paper. Also cite this work: https://www.medrxiv.org/content/10.1101/2020.06.25.20139725v6 which is a pre-print of a paper that performs a GWAS on vascular tortuosity in the UK Biobank. Of note this work is still in pre-print only, however.

Page 6 lin 223: COL4A1 (next to COL4A2) is the gene thought to be causal of the medialian disorders FRAT and HANAC. I’m wondering why FRAT is not mentioned in this paper (HANAC is less well known but also thought to be caused by COL4A1 mutation). FRAT is a well known genetic cause of vascular tortuosity. It’s interesting that COL4A2, the neighboring gene, comes up – perhaps the authors can comment on this – are we wrong about COL4A1 being the cause of FRAT? Or is the authors’ signal actually tagging COL4A1? How are COL4A1 and COL4A2 related intracellularly?

Page 7 line 247: Great that the authors replicate their work in this paper. However, the text should be more specific. They report a “generally linear relationship” between effect sizes – please report the specific correlation with confidence interval and p-value, which should be strong based on the supplementary figure.

Page 7 line 258: report p-value and confidence interval for correlations with other ancestries

Page 7 line 280: can the authors be more statistically specific – what is the exact p-value, how many SNPs/eQTLs were tested, what is the resulting Bonferroni cutoff.

Page 8 line 293: The discussing of two-sample MR analysis to assess causality is really interesting. Yet, certain clinical observations make it hard to reconcile that retinal arteriolar tortuosity causes elevated diastolic blood pressure. A mechanical elevation in systemic blood pressure via microvasculature is more believable in the kidneys, for example, where there is a large enough tissue mass to actually affect systemic blood pressure. But the retinal arteries are massively smaller in quantity. Can the authors please provide more details as to how well powered they were to decide about causality direction? Could there be an unmeasured confounder here? For example, in the situation of RVO (2% prevalence) or impending RVO, we know that vascular tortuosity comes second, with elevated blood pressure being in the causal pathway. On longitudinal patient exam, the vessels are normal, then there is uncontrolled hypertension which leads to thickening of arteriolar walls and subsequent compression of the vein, resulting in venous tortuosity with associated RVO or impending RVO. To me, this seems like the opposite direct as is described by the authors.

**Have all data underlying the figures and results presented in the manuscript been provided?**

Reviewer #1: Yes

Reviewer #2: Yes

PLOS authors have the option to publish the peer review history of their article (what does this mean?). If published, this will include your full peer review and any attached files.

Reviewer #1: No

Reviewer #2: No

---

## [Decision Letter · Decision Letter 1]

20 Dec 2022

Dear Dr Owen,

We are pleased to inform you that your manuscript entitled "GWAS on retinal vasculometry phenotypes" has been editorially accepted for publication in PLOS Genetics. Congratulations!

Yours sincerely,

Sudha K Iyengar

Academic Editor

PLOS Genetics

Hua Tang

Section Editor

PLOS Genetics

Comments from the reviewers (if applicable):

Reviewer's Responses to Questions

**Comments to the Authors:**

Reviewer #1: The authors have responded satisfactorily to my previous comments.

Reviewer #2: Thank you to the authors for your detailed responses. All my questions were answered.

**Have all data underlying the figures and results presented in the manuscript been provided?**

Reviewer #1: Yes

Reviewer #2: Yes

PLOS authors have the option to publish the peer review history of their article (what does this mean?). If published, this will include your full peer review and any attached files.

Reviewer #1: No

Reviewer #2: No

**Data Deposition**

http://datadryad.org/submit?journalID=pgenetics&manu=PGENETICS-D-22-00757R1

**Press Queries**

---

## [Editor Report · Acceptance letter]

19 Jan 2023

PGENETICS-D-22-00757R1 

GWAS on retinal vasculometry phenotypes 

Dear Dr Owen, 

We are pleased to inform you that your manuscript entitled "GWAS on retinal vasculometry phenotypes" has been formally accepted for publication in PLOS Genetics! Your manuscript is now with our production department and you will be notified of the publication date in due course.

With kind regards,

Marianna Bach

PLOS Genetics

On behalf of:
